# Deformation Mechanisms in Ni-Based Superalloys at Room and Elevated Temperatures Studied by In Situ Neutron Diffraction and Electron Microscopy

**Frank Kümmel** [1,*] , **Andreas Kirchmayer** [2] , **Cecilia Solís** [1,3] , **Michael Hofmann** [1] , **Steffen Neumeier** [2] and **Ralph Gilles** [1]

1   Heinz Maier-Leibnitz Zentrum (MLZ), TU München, Lichtenbergstr. 1, 85748 Garching, Germany; cecilia.solis@hzg.de (C.S.); Michael.Hofmann@frm2.tum.de (M.H.); Ralph.Gilles@frm2.tum.de (R.G.)
2   Department of Materials Science & Engineering, Institute I: General Materials Properties, Friedrich-Alexander-Universität Erlangen-Nürnberg (FAU), Martensstr. 5, 91058 Erlangen, Germany; andreas.kirchmayer@fau.de (A.K.); steffen.neumeier@fau.de (S.N.)
3   German Engineering Materials Science Centre (GEMS) at Heinz Maier-Leibnitz Zentrum (MLZ), Helmholtz-Zentrum Geesthacht GmbH, Lichtenbergstr. 1, 85748 Garching bei München, Germany
\*   Correspondence: frank.kuemmel@frm2.tum.de; Tel.: +49-89289-54827

**Abstract:** Polycrystalline Ni-based superalloys are one of the most frequently used materials for high temperature load-bearing applications due to their superior mechanical strength and chemical resistance. In this paper, we presented an in situ diffraction study on the tensile deformation behavior of the polycrystalline Ni-based superalloy VDM® Alloy 780 at temperatures up to 500 °C performed at the STRESS-SPEC neutron diffractometer at the Heinz Maier-Leibnitz Zentrum. A detailed microstructural investigation was carried out by electron microscopy before and after testing. The results of these studies allowed us to determine the deformation mechanism in the differently oriented grains. It is shown that the deformation behavior, which is mainly dislocation motion and shearing of the $\gamma'$-precipitates, does not change at this temperature range. The deformation is strongly anisotropic and depends on the grain orientation. The macroscopic hardening can mainly be attributed to plastic deformation in grains, where the (200) lattice planes were oriented perpendicular to the loading direction. Accordingly, a remaining lattice strain and high dislocation density were detected predominantly in these grains.

**Keywords:** Ni-based superalloy; deformation behavior; neutron diffraction; lattice strain; microstructural investigation; transmission electron microscopy

## 1. Introduction

Polycrystalline Ni-based superalloys are the most frequently used materials for structural applications such as casings or highly stressed rotating disks within the hot sections of modern gas turbine engines and power generation applications [1,2]. The material requirements for these parts are very high and challenging. The most important properties are high strength, creep resistance, and good low cycle fatigue behavior at temperatures up to 750 °C. Furthermore, as these parts are typically forged, a good formability and a high oxidation resistance are required. The superior high temperature mechanical strength of these alloys can largely be attributed to the characteristic two-phase microstructure, consisting of nano-sized intermetallic precipitates ($\gamma'$, Ni3Al-based, L12 crystal structure or $\gamma''$, $Ni_3Nb$-based, D0$_{22}$ crystal structure) distributed coherently within a matrix ($\gamma$, Ni-based, A1 crystal structure). Among the polycrystalline Ni-based superalloys, Alloy 718 is one of the most widely used due to its excellent combination of mechanical properties and processing characteristics [3–5]. However, the service temperature of Alloy 718 is limited to around 650 °C, because of the instability of the existing $\gamma''$-phase in this alloy, which transforms into the δ-phase, leading to a dramatic loss in the mechanical strength [5]. Many

efforts have been put into the research of new polycrystalline Ni-based alloys with operating temperatures above 650 °C, while keeping the advantageous processing characteristics of Alloy 718 during the last few decades [6,7]. One very promising candidate is the newly developed VDM® Alloy 780 (VDM 780) [8,9]. The most important differences between these two alloys are essentially the replacement of Fe by Co and a higher Al-content in combination with a lower Ti-content in VDM 780.

Neutron methods like (1) neutron diffraction (ND), (2) small angle neutron scattering (SANS), and (3) neutron imaging (NI) are well suited to study the microstructure of such superalloys [10]. These neutron methods enable the identification of the (1) present phases, their lattice parameters and volume fractions [11,12], (2) morphology and volume fraction of nano-sized precipitates [13,14], and (3) spatial representation of features such as cracks or air channels [15]. Neutrons facilitate the investigation of real bulk samples, typically several cubic centimeters, because of their high penetration depth and spot size [10,11,16]. In situ ND tensile tests are a well-established method to correlate the macroscopic material behavior to the deformation behavior in the crystallographically differently oriented grains under various loading conditions or elevated temperatures [17–21]. The ND method is based on Bragg's law (1), where $\lambda$ is the incident wavelength of the radiation, $2\theta_{(hkl)}$ is the diffraction angle, and $d_{(hkl)}$ is the lattice spacing of the (hkl) lattice plane. Changes in lattice spacing can be directly related to the total strain $\varepsilon_{tot,(hkl)}$ of the (hkl) lattice plane with the knowledge of the spacing of the undeformed lattice planes $d_{0,(hkl)}$ (2). The lattice strains can also be calculated by the position of the diffraction angle $2\theta_{(hkl)}$ in a diffraction experiment (2). Different plastic deformation of grains with a particular orientation leads to a deviation in the measured lattice strain from the linear-elastic behavior [22,23]. This deviation is described as an intergranular microstrain. These microstrains can develop between adjacent grains either of the same phase due to the anisotropic elastic–plastic properties in different crystallographic planes (intergranular microstrains), or between different phases (interphase microstrains) during external loading [22,24,25]. The intergranular strain in a specific (hkl) lattice plane $\varepsilon_{intergranular,(hkl)}$ can be calculated with Equation (3), where $\sigma$ is the applied stress and $E_{hkl}$ is the elastic constant of the lattice plane (hkl).

$$\lambda = 2d_{(hkl)} \cdot sin\theta_{(hkl)} \tag{1}$$

$$\varepsilon_{tot,(hkl)} = \frac{d_{(hkl)} - d_{0,(hkl)}}{d_{0,(hkl)}} = \frac{sin\,\theta_{0,(hkl)}}{sin\,\theta_{(hkl)}} - 1 \tag{2}$$

$$\varepsilon_{intergranular,(hkl)} = \varepsilon_{tot,(hkl)} - \frac{\sigma}{E_{hkl}} \tag{3}$$

The microstructure at the room and elevated temperature of VDM 780 has already been investigated in previous publications by the authors of this paper by scanning electron microscopy (SEM), high-resolution transmission electron microscopy (HRTEM) as well as ND and SANS [26–28]. The microstructure consists of $\gamma$-matrix, hardening $\gamma'$-phase, and high temperature stable $\delta$- and $\eta$-phases. Their fractions strongly depend on the underlying annealing treatments. However, the behavior of these phases under mechanical load has not been investigated yet. In this study, in situ ND experiments were performed on bulk VDM 780 samples to investigate the material behavior under uniaxial tensile deformation at different temperatures. The microstructure of the samples was analyzed before and after the tensile tests by SEM and TEM to determine the deformation mechanism.

## 2. Materials and Methods

The chemical composition of the investigated Ni-based VDM 780 is depicted in Table 1. In this study, the samples were heat-treated to obtain a microstructure mainly composed of $\gamma$ and $\gamma'$ (1020 °C/1 h/air cooled + 980 °C/1.5 h/water cooled + 720 °C/8 h/furnace cooled with 50 °C/h to 620 °C + 620 °C/8 h/air cooled). Previous ND, SANS, and atom probe investigations have shown that this heat treatment leads to a bimodal size distribution of the $\gamma'$-precipitates with a mean diameter of 13 (17 vol.%) and 5 nm (4 vol.%). Furthermore,

Nb-based carbides/nitrides were also observed with a fraction of approximately 1 vol.%. These results will be published in more detail in a future paper.

**Table 1.** Chemical composition of the investigated VDM® Alloy 780 samples as measured by spark spectrometry.

| Element | Co | Cr | Nb | Mo | Al | Fe | Ti | Ni |
|---------|------|------|-----|-----|-----|-----|-----|------|
| wt.% | 24.4 | 17.7 | 5.4 | 2.9 | 2.2 | 0.6 | 0.3 | Bal. |

The in situ neutron diffraction measurements were carried out at the materials science diffractometer STRESS-SPEC [29,30] of the Research Neutron Source Heinz Maier-Leibnitz (FRM II) in Germany. STRESS-SPEC was optimized for the determination of strains and textures in crystalline materials due to the variable wavelength and the possibility of measuring with small gauge volumes at scattering angles close to 90°. The high flux of the neutron source additionally allows useful measurement statistics in reasonable time periods. The detector with its $25 \times 25$ cm$^2$ area covered a range of $\Delta 2\theta = 5$–$20°$, depending on the distance between the sample and detector. A Ge (311) monochromator was used to set a constant wavelength of approximately 1.65 Å in the current tests. The angular resolution was enhanced by an 'in-pile' collimator with a horizontal divergence of 25'. The size of the beam in front of the sample was set with high precision slits to $5 \times 5$ mm$^2$. The final horizontal dimension of the gauge volume was defined by a radial collimator with a full width at half maximum (FWHM) of 5 mm in the diffracted beam. The measurement direction, which is the normal direction to the diffracting (hkl) lattice planes, was parallel to the applied load for all samples. A schematic of the experimental setup is depicted in Figure 1. The measuring time was 120 s for the 2θ-range 44–58°, where (111) and (200) $\gamma/\gamma'$ peaks are expected, and 600 s for the 2θ-ranges 73–87°, (220) $\gamma/\gamma'$, and 91–105°, (311) $\gamma/\gamma'$. A single-peak analysis of the (111), (200), (220), and (311) peaks was performed to extract the position, intensity, and FWHM with the software StressTextureCalculator [31,32]. A Gaussian function was used to fit the profile shape.

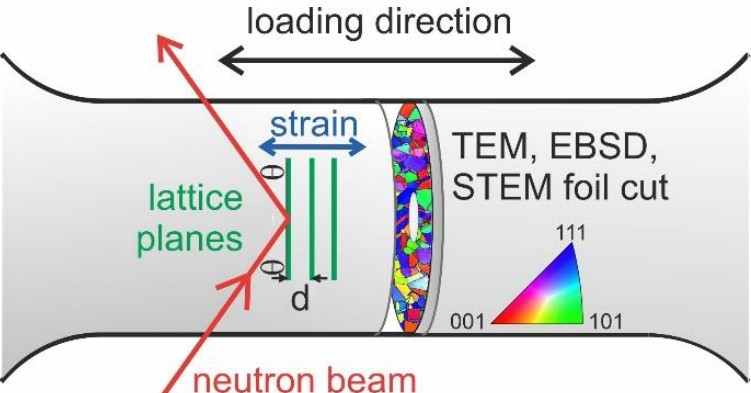

**Figure 1.** Schematic view of the investigated lattice planes by the neutron beam in relation to the direction of loading and introduced lattice strain of the tensile specimen. The thin electron transparent foils that were taken perpendicular to the loading direction were characterized microscopically using Transmission electron microscopy (TEM), Electron backscatter diffraction (EBSD), and scanning transmission electron microscope (STEM).

The tensile tests were performed using a testing machine developed in-house (Figure 2). This testing machine was optimized to simulate the mechanical and thermal conditions during material processing as well as during the operating conditions of high-strength materials such as Ni-based superalloys. Heating was provided via a direct current resistivity method and the sample temperature was measured by a spot-welded thermocouple. The specimen extension was determined by a EXH10-5A high temperature extensometer (Sand-

ner Messtechnik GmbH, Biebesheim, Germany), which was connected to the sample by two ceramic rods. The specimen was heated up to the respective testing temperatures and afterward, the displacement was stepwise increased during the current ND experiments. The position was kept constant for about 45 min at each step while the diffraction patterns were recorded. The samples were unloaded in position control at a total strain of about 8% to measure the elastic properties of the material.

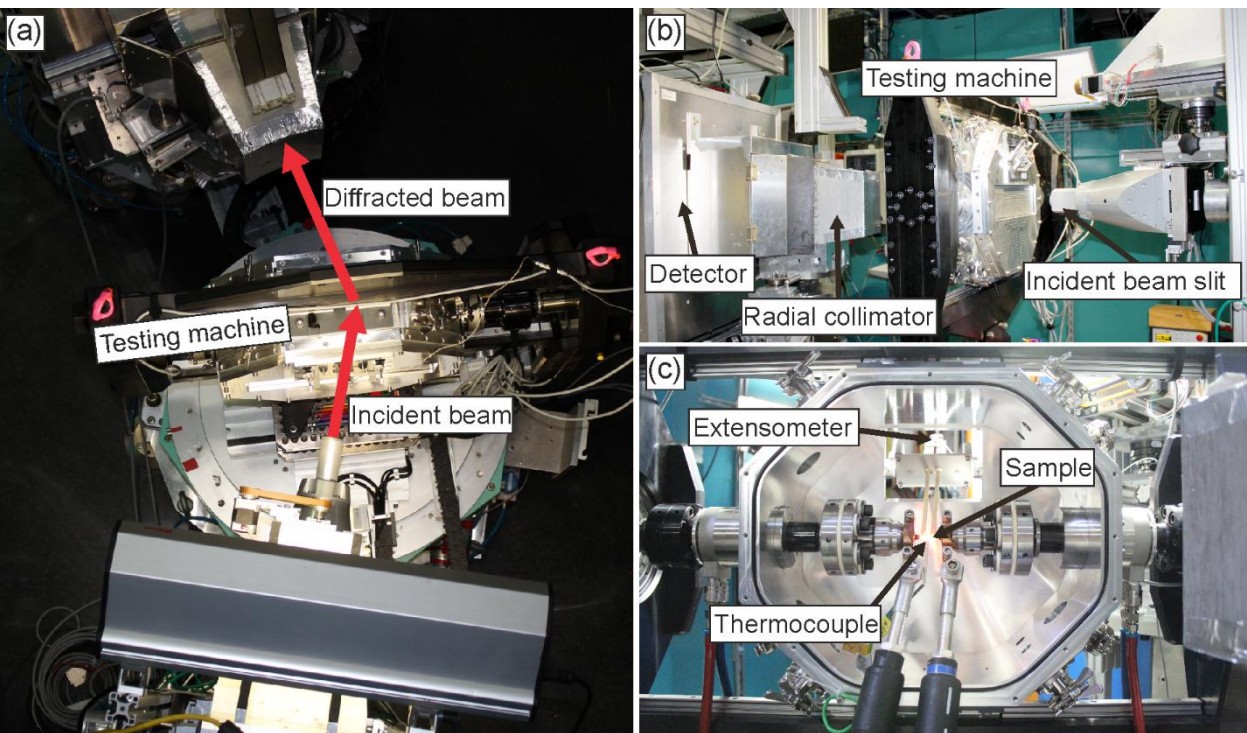

**Figure 2.** Overview of the experimental setup at the STRESS-SPEC instrument in (**a**) top, (**b**) side-view and (**c**) detail view into the heating chamber of the testing machine.

The microstructure of the samples was investigated by SEM and TEM before and after the tensile tests to determine the deformation mechanism. Backscattered electron (BSE) images were made using a Zeiss Crossbeam XB540 SEM (Carl Zeiss AG, Oberkochen, Germany) with a working distance of 8.5 mm and an acceleration voltage of 20 kV. Electron backscattered diffraction (EBSD) measurements were made with the same SEM at a working distance of 16 mm and 20 kV acceleration voltage using a Nordlyn II detector (Oxford Instruments plc, Abingdon, UK). The software used to analyze the EBSD data were Aztec and HKL Channel 5 (Oxford Instruments). Samples for both characterization methods were taken perpendicularly to the loading axis from the deformed gauge length and from the undeformed screw thread to evaluate the change in the microstructure. The samples were mechanically ground and polished with an OP-U active oxide suspension (Struers Aps, Ballerup, Denmark). TEM samples were taken from the gauge length perpendicularly to the loading axis, mechanically ground, and afterward electrochemically further thinned using an electrolyte solution containing 8% perchloric acid and an applied voltage of 40 V at a temperature of $-30\ °C$. Two beam bright and dark field (BF/DF) images were recorded on a Philips CM200 TEM, operating with an acceleration voltage of 200 kV and using the installed 2k × 2k CMOS camera F216 (Tietz Video and Image Processing Systems GmbH, Gauting, Germany). On the same electron transparent samples, EBSD and STEM investigations were conducted with a Zeiss Crossbeam XB540 SEM. For the latter, an acceleration voltage of 30 kV and a working distance of 2 mm and a detector distance of 9 mm was used. BF and high angular annular dark field (HAADF) images were taken.

## 3. Results and Discussion

### 3.1. In Situ Neutron Diffraction Studies

The macroscopic stress-strain curves of the VDM 780 specimens at 25 and 500 °C are displayed in Figure 3. The symbols indicate points at which the strain was held constant for about 45 min and the diffraction patterns were recorded. No significant change in load was measured during the holding segments. Therefore, creep deformation did not play a notable role during these experiments. The heat treatment, which was performed before the in situ tensile test, did not lead to a fully hardened condition. In the VDM® Alloy 780, a tensile strength of greater than 1400 MPa can be achieved. However, the material keeps its good mechanical strength up to 500 °C and continuously hardens until unloading the sample at both temperatures in the investigated condition. The stress response at 500 °C was only about 80 MPa below the stress response at 25 °C at the same engineering strain.

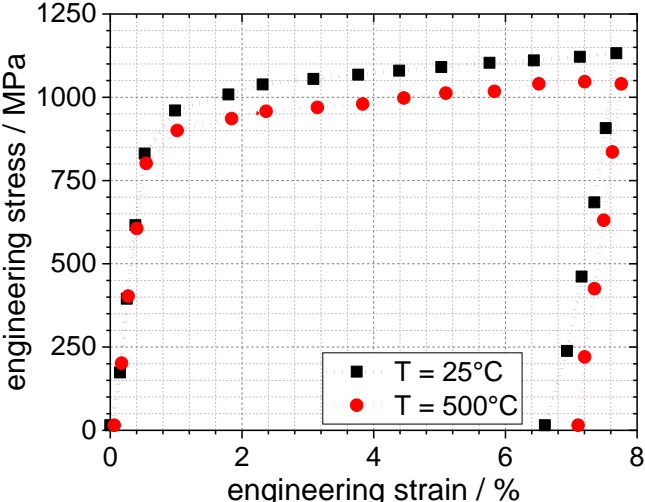

**Figure 3.** Macroscopic engineering stress/strain curves of the Ni-based VDM Alloy 780 at 25 and 500 °C. The symbols indicate the points of constant strain holding while the diffraction patterns were taken.

The diffraction patterns are shown in Figure 4 at selected specimen strains (at the beginning and the end of the elastic regime as well as the middle and at the end of the plastic regime) for the testing temperatures (a) 25 °C and (b) 500 °C. The shape and position of the Bragg peaks change with increasing applied load. The peak intensity and FWHM are the most important parameters that describe the peak. A change in the peak intensity can be correlated to a change in the preferred orientation of a phase. A change in the FWHM can be related to a change in the material defects like crystallite size or microstrains [33,34]. However, the crystallite size has to be rather small, usually clearly below 1 µm, to effectively influence the peak broadness [35,36]. Differences in the measured FWHM can therefore be related to a change in the microstrain that stems from a rising density of structural defects like dislocations or stacking faults in the grains, as the grain size is around 20 µm in this study. The results of the Gaussian curve fitting of the individual peaks are exemplarily shown for the sample tested at 25 °C in the unloaded state ($\sigma = 0$ MPa) in Figure 4c. The shape of the peaks was asymmetric. The broad peak shoulder on the left side can be attributed to the $\gamma'$-phase [26]. However, these additional peaks did not notably influence the peak height and broadness of the main peak, as the fraction of the $\gamma'$-phase was significantly lower than the one of the $\gamma$-phase, and the lattice misfit of 0.48 in between these two phases is rather high in this alloy [26]. Furthermore, the fitting procedure was focused on obtaining a good fit quality at the right bottom side as well as the peak width and height of the main peak. The Gaussian curve fits the position, intensity, and FWHM of the primary peak ($\gamma$-phase) quite well. The only major differences between the measured

data and the fitting curves were found at the left side of the primary peaks, at which the peaks of the γ′-phase occur.

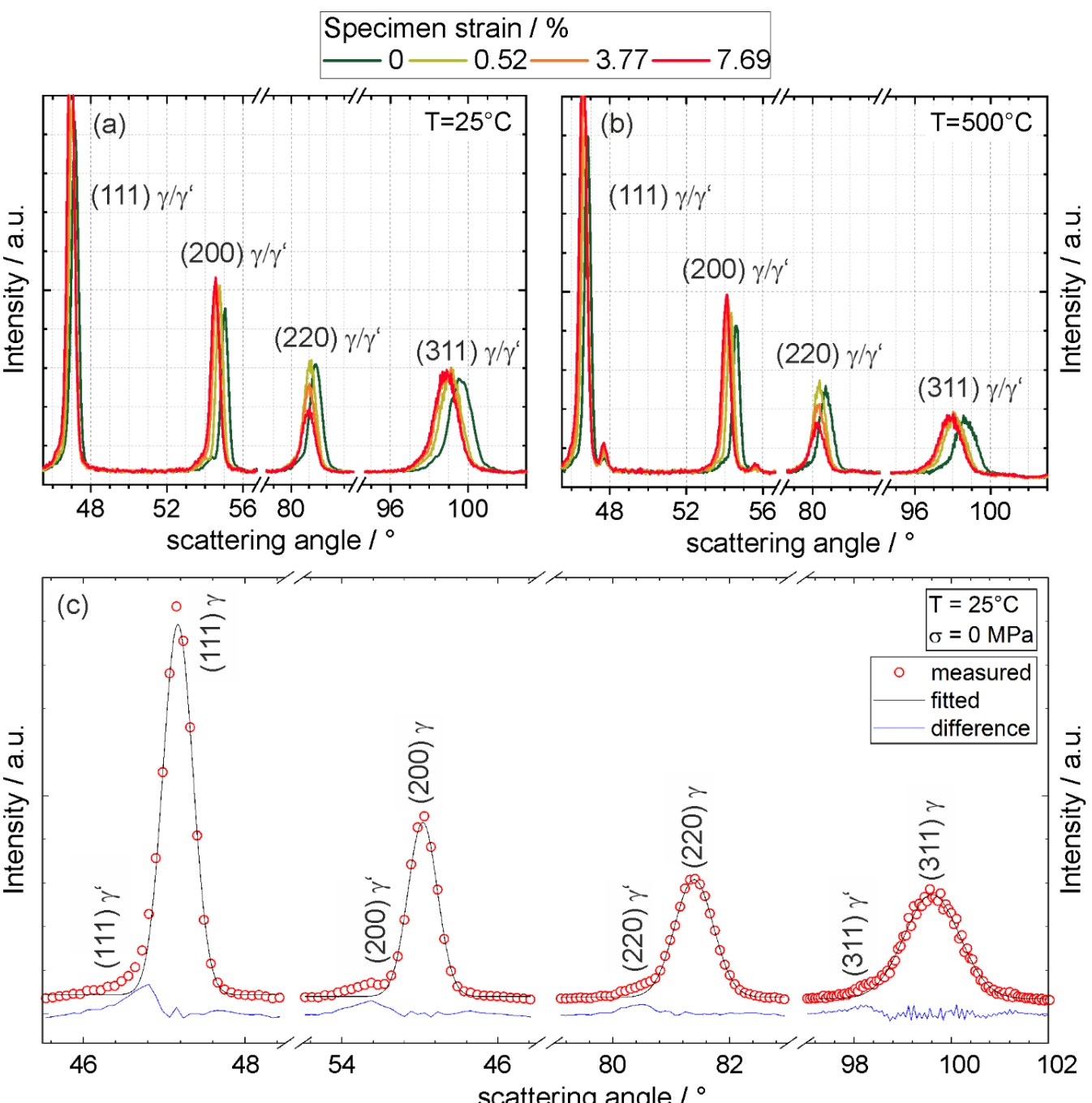

**Figure 4.** Neutron diffraction pattern at different specimen strains (0, 0.52, 3.77, and 7.69%) performed at (**a**) 25 °C and (**b**) 500 °C. The results of the Gaussian curve fitting of the individual peaks is exemplarily shown for the sample tested at 25 °C in the unloaded state (σ = 0 MPa) in (**c**).

The development of the peak intensity and lattice strain with increasing applied load are shown in Figure 5. The peak intensity and FWHM were normalized to the value of the unloaded sample at the beginning of the experiment in these diagrams to compensate for the different initial values.

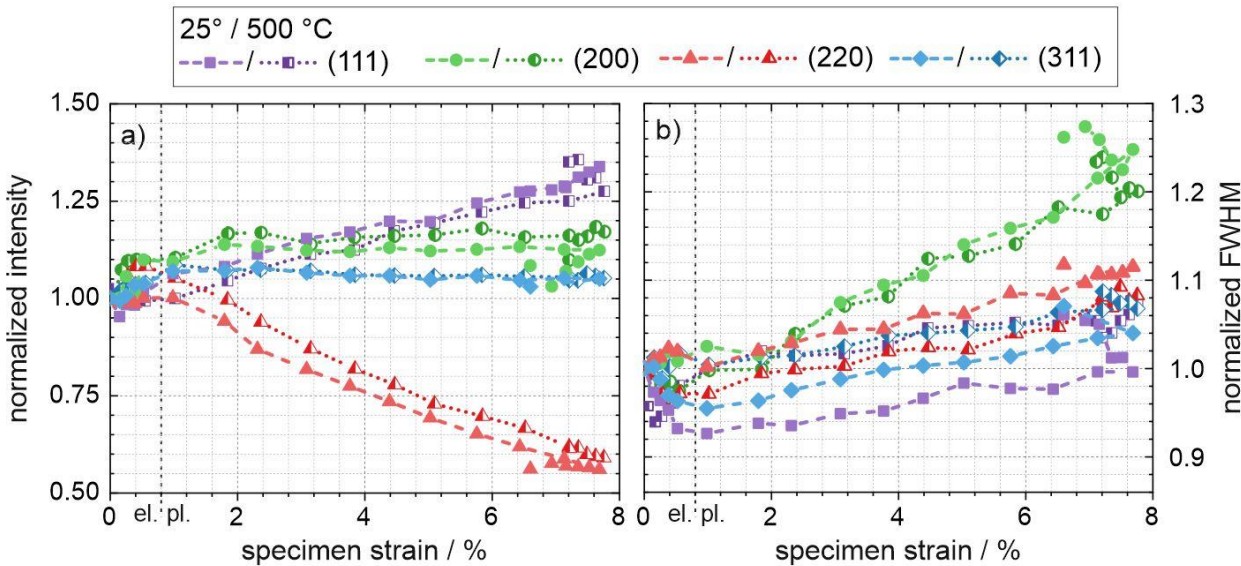

**Figure 5.** (**a**) Normalized peak intensity and (**b**) normalized full width at half maximum (FWHM) of peaks in different crystallographic orientations depending on the specimen (engineering) strain.

The peak intensity change was similar for both tested temperatures (Figure 5a). The intensity of the (111), (200), and (220) peaks increased during the deformation experiment. The (311) peak exhibited the lowest change in intensity. The intensity increased until the beginning of the macroscopic plastic regime stayed approximately at this value during further plastic deformation. The (200) peak revealed a similar behavior at a marginally higher value. In contrast, the intensity of the (111) peak increased linearly during the whole experiment up to a considerably higher value. It was particularly noticeable that the intensity of the (220) peak strongly decreased during the plastic deformation. These findings are similar to the literature data for other Ni-based alloys [17,33]. The re-orientation is well known for polycrystalline metals, whereas the deformation resistance differs in the crystallographic orientations. Grains in face centered cubic materials, like Ni-based alloys, that are not favorably oriented for deformation tend to rotate in order to activate deformation on the preferred slip systems [37].

The development of the FWHM in the different peaks with increasing sample strain was also similar for both temperatures (Figure 5b). The FWHM of the (111) and (311) peak slightly decreased while the one of the (200) and (220) peak stayed almost constant during the elastic regime. In contrast, the FWHM continuously increased with rising plastic deformation. The increase in FWHM was correlated to a rising internal strain, which was probably caused by an increasing dislocation density [38]. The FWHM of all peaks further increased during unloading. The maximum values were strongly dependent on the crystallographic orientation. The maximum increase from the lowest FHWM to the highest FHWM was very similar in the (111), (220), and (311) peaks. The FWHM increased linearly by about 10% during plastic deformation in these lattice planes. The biggest changes were present for the (200) peak, at which the FWHM increased more than 20% at both temperatures. The decreasing FWHM with increasing elastic strain and the increasing FWHM during the unloading segment at the end of the experiment is an unusual material behavior. This can only be explained by internal stresses in the unloaded sample resulting from geometrical necessary dislocations. These dislocations can arise at two different sites: (1) the $\gamma/\gamma'$-phase boundary because of the different lattice parameter and thermal expansion coefficient [26] or (2) the $\gamma/\gamma$ grain boundary, at which dislocation can pile up during deformation and which can act as a sink and source for dislocations [39,40]. It is known in the literature [41] that these dislocations are rearranged, if the applied stress on the samples is changed. This leads to a reduction in the dislocation density in some lattice planes at the beginning of the loading in the current experiments, whereas almost no new

dislocations are generated, whereas the opposite behavior takes place during unloading. This effect was most pronounced in the (111) lattice planes as the Young's modulus in this orientation was the highest (Table 2), leading to the highest internal stresses.

The influence of the applied specimen strain on the total lattice strain and on the intergranular strains is shown in Figure 6a,b, respectively. The strains were calculated according to Equations (2) and (3), respectively. The deformation in all lattice planes followed the same trend at 25 and 500 °C. The lattice strains were only slightly higher at 500 °C. The strain was rather anisotropic in the different lattice planes leading to significant intergranular strains (differences between lattice strains measured on different (hkl)-planes). The total strain in the (111) and (220) lattice planes was the lowest during the whole experiment. The (311) lattice planes showed higher lattice strains compared to (111) and (220). However, this difference was mainly due to the different Young's moduli of the lattice planes (Table 2). The intergranular strain was almost equal in the (111) and (311) lattice planes (Figure 6b), and almost no intergranular strain remained in these lattice planes after the experiment. The largest intergranular strain was found in the (200) lattice planes during the whole experiment, which resulted in a considerably higher remaining strain in the (200) lattice planes. The grains in which the (200) lattice planes were perpendicular to the loading direction were the most favorably orientated ones for dislocation activities in the preferred {111} <110> slip systems. This facilitates plastic deformation at considerably lower stresses in these grains [37,42]. It is important to point out that a negative intergranular strain was measured in the (220) lattice planes, which means that the grains that are oriented in the [220] direction are compressed during the plastic deformation in the loading direction. The explanation of the opposite behavior in the (220) lattice planes has to be based on consolidation mechanisms between adjunct grains and the formation of textures during plastic deformation [43].

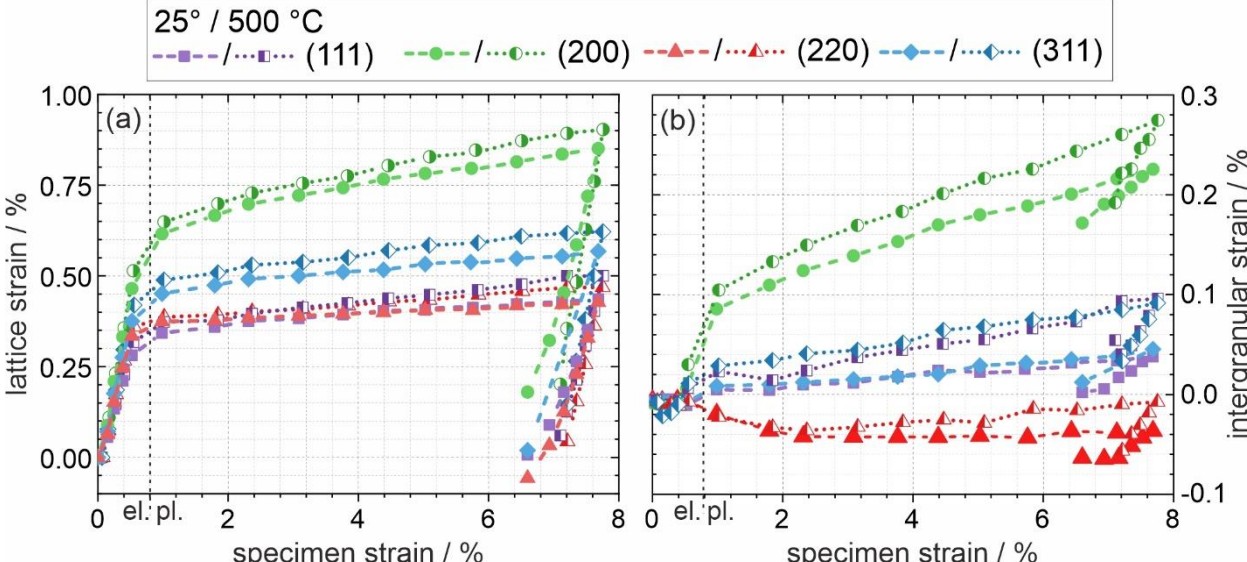

**Figure 6.** (**a**) Total lattice and (**b**) intergranular strain in the lattice planes with different crystallographic orientations depending on the specimen (engineering) strain.

**Table 2.** Overview of the material properties in different lattice planes determined by neutron diffraction at 25 °C.

| | (111) | (200) | (220) | (311) |
|---|---|---|---|---|
| Lattice plane |  |  |  |  |
| Number of atoms in lattice plane | Very high  | Medium  | Low  | Medium  |
| Young's Modulus | Very high (256 GPa) | Low (167 GPa) | High (228 GPa) | Medium (204 GPa) |
| Rel. change in intensity (0.52%/7.69%) | Very high (1%/33%) | High (10%/12%) | Negative (0%/−44%) | Low (4%/5%) |
| Rel. change in FWHM (0.52%/7.69%) | Low (−7%/1%) | Very high (1%/25%) | High (2%/12%) | Medium (−4%/4%) |
| Lattice strain (0.52%/7.69%) | Low (0.3%/0.4%) | Very high (0.5%/0.9%) | Low (0.3%/0.4%) | Medium (0.4%/0.6%) |

### 3.2. Microstructural Analysis of the Deformation Behavior

The in situ ND experiments have demonstrated that the macroscopic deformation and the hardening behavior in the plastic deformation regime were very similar at both temperatures. The only difference was the lower yield strength at 500 °C. The deformation behavior in the different lattice planes was also very similar for both temperatures, which can be seen by the almost identical behavior of the FWHM and the lattice strains. This confirms that the deformation behavior does not change within the tested temperature range in this alloy.

A detailed microstructural characterization was performed for the samples deformed at 25 and 500 °C and very similar deformation mechanisms were found at both temperatures. Selected microstructures of the undeformed and deformed areas of samples tested at room temperature or 500 °C are depicted in this section. Figure 7a,d depicts the deformed and undeformed microstructure in BSE contrast. It is apparent from the blurred regions in Figure 7d that the deformation of about 8% specimen strain leads to high lattice distortion in some areas. However, compared to Figure 7b,e, the deformation seems to have no significant influence on the grain structure. The median grain size of the undeformed and the deformed specimen was very similar at 19 and 22 µm, respectively. Furthermore, a pronounced deformation driven twinning can be excluded, since both samples showed a similar amount of twins.

The local misorientation in the undeformed and deformed sample is presented in Figure 7c,f. It is obvious that the deformation concentrated in specific areas and at grain boundaries. Macroscopic plastic deformation begins as soon as the applied stress exceeds the yield point of the material and massive dislocation movement takes place. However, the plastic deformation is not distributed homogeneously in polycrystalline materials. In these materials, the direction of the individual grains in relation to the direction of loading has to be taken into account. A dislocation moves as a result of a force that acts on it in the slip plane in the direction of the Burgers vector (slip direction). Therefore, the resulting shear stress $\tau$ in the sliding system is relevant and not the applied tensile stress $\sigma$. The resulting shear stress $\tau$ can be calculated by the tensile stress $\sigma$, the angle between the

tensile direction and direction normal to the glide plane $\varphi$, and the angle between the tensile direction and the direction of the glide plane $\omega$, Equation (4). The angle dependence can be summarized by the Schmid factor $S$.

$$\tau = \sigma \cdot \cos \varphi \cdot \cos \omega = \sigma \cdot S \tag{4}$$

During plastic deformation, suitable oriented grains with high Schmid factors deform first, however, the grains are connected and cannot deform independently from each other. Accordingly, they exert stresses on other less favorably oriented grains. To keep the cohesion of the constituent grains, geometrically necessary dislocations mainly near the grain boundaries have to form, at which the incompatibility stresses between the neighboring grains are the highest.

TEM and STEM investigations were undertaken on the deformed material to get a better insight into the deformation mechanisms in the differently oriented grains. In Figure 8, images of differently orientated grains are shown in bright field TEM condition and STEM dark field. Coupled pair-dislocations, which shear through the $\gamma/\gamma'$ microstructure due to the relatively small size of the $\gamma'$-precipitates [33,44,45] as well as stacking faults, are present in this grain orientation. No quantitative analysis of the different types of defects is provided here. Depending on the grain orientation, various shearing mechanisms can occur, as demonstrated recently by León-Cázares et al. [46].

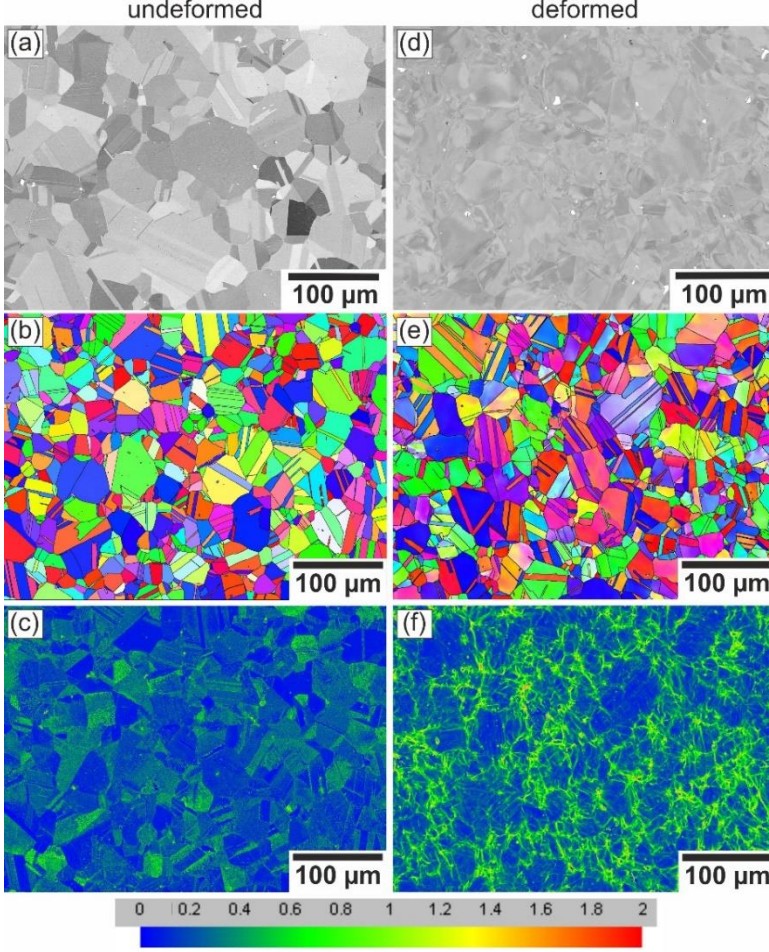

**Figure 7.** Comparison of the (**a**) BSE and (**b**) EBSD images of the undeformed specimen with the (**d**) BSE and (**e**) EBSD images of the deformed microstructure of the specimen measured at room temperature. (**c**,**f**) show the local misorientation of the undeformed and deformed specimen with a threshold of 3 degrees.

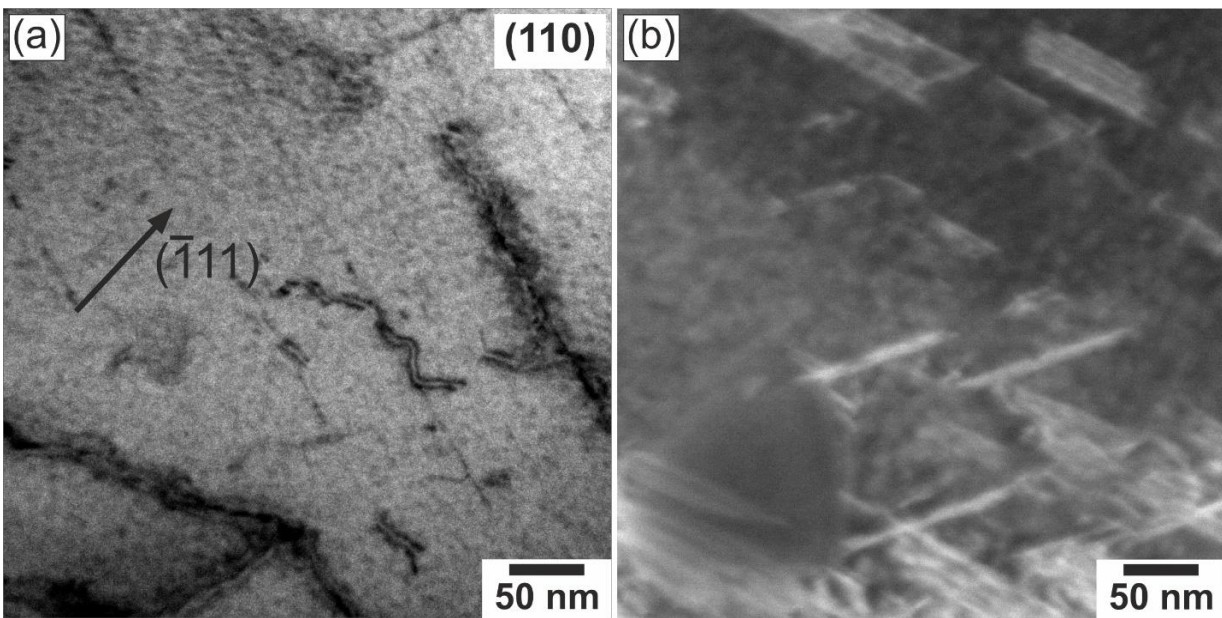

**Figure 8.** (**a**) Bright field TEM image of the sample deformed at room temperature with dislocations in a <110> oriented grain viewed in two-beam condition close to (110) and (**b**) STEM dark field images of the sample deformed at 500 °C.

Additionally, STEM analysis was performed on a TEM foil. Figure 9a–f show BF and HAADF STEM images of grains in different orientations. The orientation was determined by EBSD (Figure 9g). Multiple active {111} <110> glide planes are found in the [002] orientated grain (Figure 9a,b). Apart from the distinct slip planes, single dislocation pairs homogenously distributed throughout the grain can be seen. In contrast, fewer dislocations and no strongly activated glide planes were visible in the grain oriented [334] (i.e., close to [111]) (Figure 9c,d). This correlates with the smaller Schmid factors of this oriented grain compared to the [002] oriented grain and indicates that less deformation takes place there. A table with all Schmid factors for the measured grains is depicted in Table 3. This confirms that the results presented in the previous chapter that the grains that are oriented in <111> were much less plastically deformed than the <200> oriented ones. The [202] orientated grain predominantly showed glide on only one slip system (Figure 9e,f), which can also be explained by the Schmid factors. Similar to the [002] orientation, dislocation activity was visible apart from the strongly activated glide planes, although less than in the [002] oriented grain.

**Table 3.** Schmid factors of the investigated grains for different slip systems calculated from Euler angles measured with EBSD with applied load in the z direction (parallel to the grain orientation).

| Slip Plane | Slip Direction | Grain Orientation | | |
|---|---|---|---|---|
| | | **[002]** | **[202]** | **[334]** |
| (111) | $[01\bar{1}]$ | 0.41 | 0.41 | 0.15 |
| | $[\bar{1}01]$ | 0.43 | 0.02 | 0.13 |
| | $[1\bar{1}0]$ | 0.03 | 0.38 | 0.02 |
| $(\bar{1}11)$ | $[0\bar{1}1]$ | 0.41 | 0 | 0.15 |
| | $[101]$ | 0.38 | 0 | 0.15 |
| | $[\bar{1}\bar{1}0]$ | 0.02 | 0 | 0 |
| $(1\bar{1}1)$ | $[0\bar{1}1]$ | 0.40 | 0.02 | 0.06 |
| | $[\bar{1}0\bar{1}]$ | 0.43 | 0.05 | 0.34 |
| | $[110]$ | 0.03 | 0.03 | 0.28 |
| $(\bar{1}1\bar{1})$ | $[0\bar{1}1]$ | 0.41 | 0.43 | 0.35 |
| | $[10\bar{1}]$ | 0.38 | 0.02 | 0.06 |
| | $[\bar{1}\bar{1}0]$ | 0.03 | 0.41 | 0.30 |

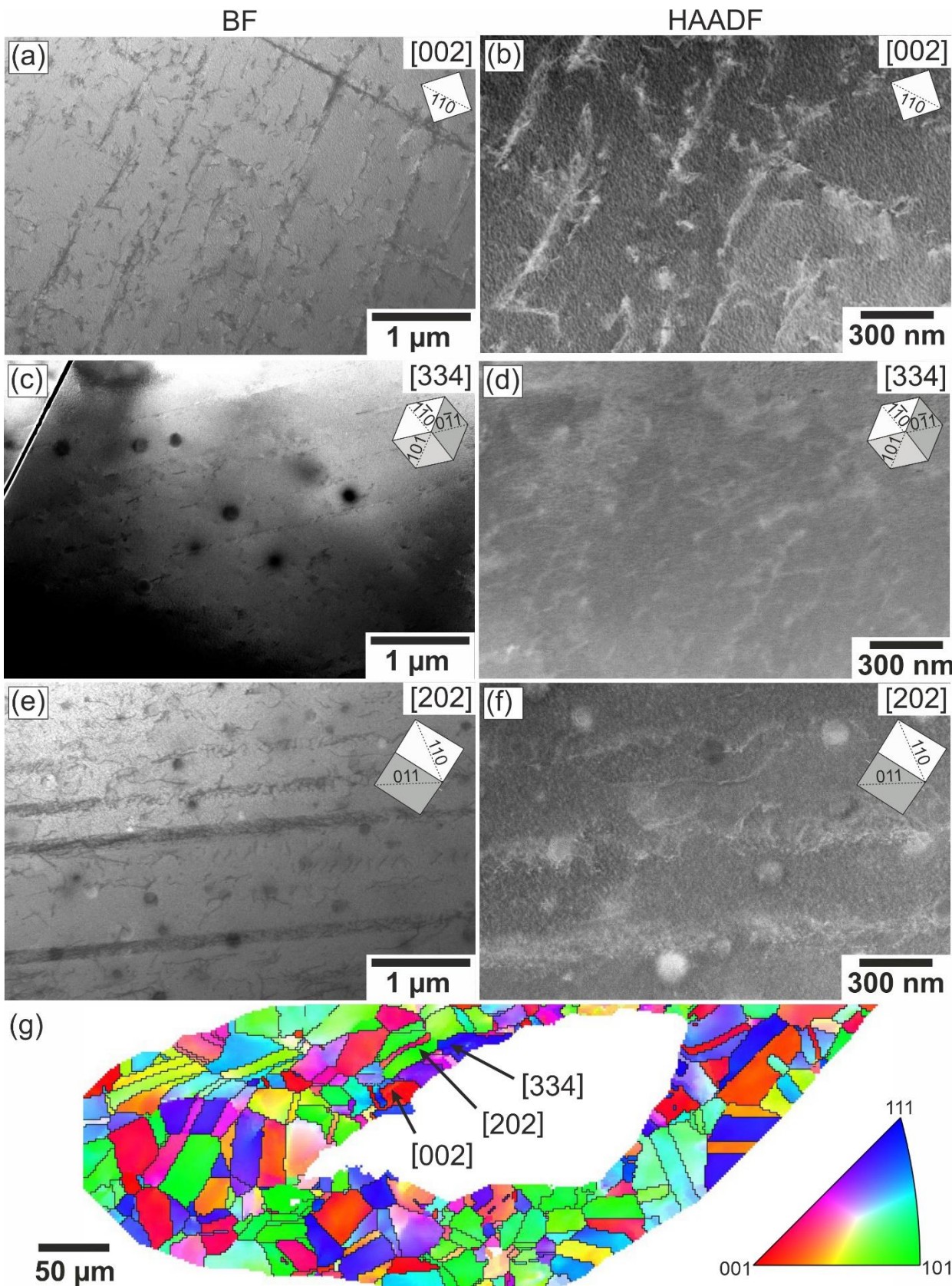

**Figure 9.** (**a**,**c**,**e**) BF and (**b**,**d**,**f**) HAADF STEM images of three grains with different orientations in the sample deformed at room temperature. The viewing direction is parallel to the loading direction. The cube in the upper right corner illustrates the crystallographic orientation of the grain. The orientation was determined by EBSD (**g**).

### 3.3. Comparison of the Deformation Behavior to the Literature Data

A comparison of the deformation behavior at micro-scale at room temperature is depicted for the VDM® Alloy 780 and Haynes® 282 [47–49] in Figure 10. Haynes 282 is a polycrystalline $\gamma'$-strengthened superalloy, which was also developed for high temperature structural applications, especially those in aero and industrial gas turbine engines. The main differences in chemical composition in comparison with VDM 780 is a lower Co-content (10 wt.%) and a higher Mo- and Ti-content (8.5/2.1 wt.%). The Haynes 282 samples in this comparative study [20] were solution annealed at 1010 °C for two hours, then water-quenched, and afterward age-hardened at 788 °C for eight hours and air-cooled. The Haynes samples had a similar microstructure as the VDM 780 samples after this heat treatment [20]. The $\gamma$-grain size was approximately 22 µm and the diameter of the $\gamma'$-precipitates was about 25 nm. However, the fraction of the $\gamma'$-precipitates in the Haynes 282 samples (15 vol.%) was lower than in the investigated VDM 780 samples (21 vol.%). Additionally, titanium nitrides and carbides with high Mo-content were observed in Haynes 282 with volume fractions below 1%, which was also similar to the VDM 780 samples. The increase in the lattice strain in the different lattice planes with rising sample strain was almost identical in both alloys (Figure 10). The major part of the plastic deformation was localized in the grains, at which the (200) lattice planes were orientated perpendicular to the loading direction. Therefore, very similar deformation mechanisms seem to be active in the two different materials, which is a result of the similar size of the $\gamma'$-precipitates. In [20], it was noted that the dominant deformation mechanism in the Haynes 282 samples was shearing of the $\gamma'$-precipitates, as a nearly identical mechanical behavior between the $\gamma$- and the $\gamma'$-phase, was observed. This is in good agreement with the microstructural investigations of VDM 780 in this work.

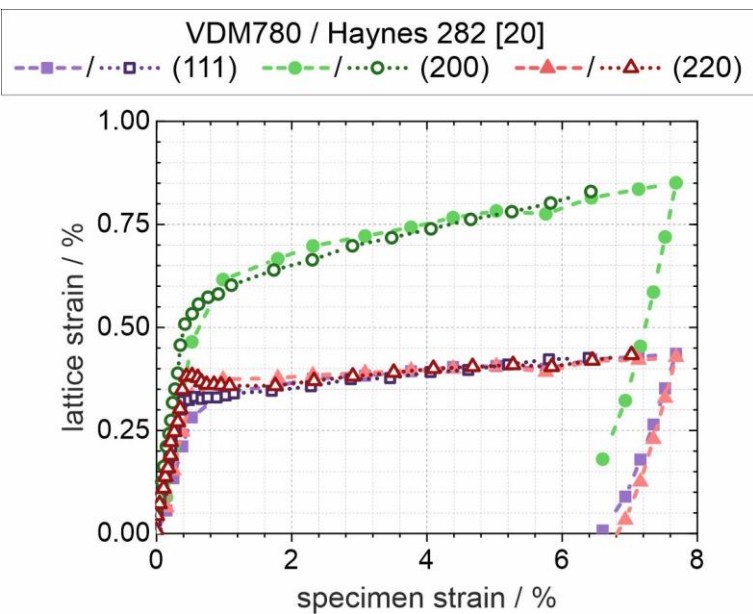

**Figure 10.** Comparison of the microscopic mechanical properties (lattice strain in different crystallographic orientations/specimen strain) of VDM® Alloy 780 (full symbols) with the literature data of the Ni-based superalloy Haynes® 282 (open symbols) [20]. The tensile test with Haynes 282 was stopped at a specimen strain of about 6.5%.

The elastic constants in the different lattice planes were determined by the loading or unloading curves. The elastic constants at 25 and 500 °C are depicted for VDM 780 in Table 4. The highest values were found for the (111), followed by the (220), (311), and (200) lattice planes. The elastic constants were significantly lower at the higher testing temperature. These values were compared to the literature data for Inconel® Alloy 718, whose primary hardening phase was $\gamma''$ (approx. 17 vol.%) [19]. The elastic constants for

VDM 780 were very similar to those of Alloy 718. Furthermore, these elastic constants were also consistent with the data measured at room temperature on the Ni-based alloys Inconel® Alloy 625 [21] or Waspalloy® [50]. It was reported in [19] that the precipitation of the $\gamma''$-phase did not change the elastic constants, as identical values were measured in the fully solution annealed and precipitation strengthened Alloy 718 sample. Similar results were also observed in a simulated $\gamma'$-strengthened Ni-based alloy, in which the volume fraction of $\gamma'$ was varied between 25 and 75% [51].

**Table 4.** Elastic constants of the individual lattice planes. As a comparison, the literature data for Alloy 718 [19] are also given.

| Lattice Plane | Young's Modulus/GPa | | | | | | | |
|---|---|---|---|---|---|---|---|---|
| | (111) | | (200) | | (220) | | (311) | |
| Temperature | 25 °C | 500 °C | 25 °C | 500 °C | 25 °C | 500 °C | 25 °C | 500 °C |
| VDM® Alloy 780 | 256 | 219 | 167 | 149 | 228 | 194 | 204 | 173 |
| Inconel® Alloy 718 [19] | 261 | 231 | 166 | 144 | 232 | 199 | 201 | 174 |

## 4. Summary and Conclusions

In situ tensile loading and unloading experiments were performed with a newly developed testing machine in a neutron diffractometer to investigate the deformation behavior in the Ni-based superalloy VDM® Alloy 780 at 25 and 500 °C. Furthermore, a detailed microstructural investigation was carried out by electron microscopy before and after testing. This enabled us to correlate the macroscopic mechanical properties with the micromechanical deformation behavior in differently oriented grains as follows:

- The deformation behavior did not change within this temperature range. At both tested temperatures, the material showed macroscopic strain hardening and the deformation mechanism was mainly shearing of the $\gamma'$-precipitates by coupled dislocation pairs and the formation of stacking faults by partial dislocations.
- The deformation was strongly anisotropic and dependent on the grain orientations. The macroscopic hardening can mainly be attributed to plastic deformation in the grains, whereas the (200) lattice planes were oriented perpendicular to the loading direction. These grains were the only ones where the peak width was strongly increased during the experiment and a remaining intergranular strain was measured after the experiment. In contrast, the grains at which the (111) lattice planes were oriented perpendicular to the loading direction displayed much less plastic deformation. This difference can be explained by the different Schmid factors.
- The elastic anisotropy of VDM 780 was similar to comparable wrought polycrystalline Ni-based superalloys. The Young's modulus was the highest in the <111> direction and the lowest in the <002> direction.

**Author Contributions:** Conceptualization: F.K., C.S., R.G.; Methodology: F.K., C.S., M.H.; Validation: F.K., C.S., R.G.; Investigation: F.K., A.K., C.S., M.H.; Data curation: F.K.; Writing—original draft preparation: F.K., A.K.; Writing–review and editing: C.S., M.H., S.N., R.G.; Supervision, project administration, and funding acquisition: S.N., R.G. All authors have read and agreed to the published version of the manuscript.

**Funding:** This research was funded by the Federal Ministry of Education and Research of Germany (project number 05K16WO2).

**Institutional Review Board Statement:** Not applicable.

**Informed Consent Statement:** Not applicable.

**Data Availability Statement:** Not applicable.

**Acknowledgments:** The authors would like to thank VDM Metals for providing the material for the tests.

**Conflicts of Interest:** The authors declare no conflict of interest. The funders had no role in the design of the study; in the collection, analyses, or interpretation of data; in the writing of the manuscript, or in the decision to publish the results.

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
