# Peer review of "Deformation Mechanisms in Ni-Based Superalloys at Room and Elevated Temperatures Studied by In Situ Neutron Diffraction and Electron Microscopy"

_metals, doi:10.3390/met11050719_

Round 1
Reviewer 1 Report
Authors study the deformation mechanisms in Ni-based superalloys at room and elevated temperatures by using in-situ neutron diffraction and electron microscopy. The data is comprehensive and solid, and the manuscript is suitable to the journal. However, I can not recommend it in current format. One of the major problem is the presentation styles and language. The paper is very hard to read and to be understood, I will expect that authors would lost too.
As a research paper, I would suggest the authors to rewrite the paper using structures as a general paper with sections: introduction, materials and methods, results, discussions and conclusions. Please keep in mind that you have clear picture to show authors your research aim, how you approach, what is new in your study, and what conclusion you draw.
For the language, because the problem exist in the entire paper. I mention here only one paragraph (conclusion parts) here for example: the entire conclusion does not really have conclusive data or results, in fact it is more like a summary. Additionally, some word like “therefore” and “furthermore” are misused in the paragraph.
I believe that the entire paper need to be rewritten to give a clear story.
Author Response
Dear reviewer,
Thank you very much for your valuable comments and suggestions on our paper. We followed all of your suggestions/comments and revised the paper accordingly. Below you will find the detailed response to your comments.
Yours sincerely,
Frank Kümmel , Andreas Kirchmayer , Cecilia Solís , Michael Hofmann , Steffen Neumeier and Ralph Gilles
Authors study the deformation mechanisms in Ni-based superalloys at room and elevated temperatures by using in-situ neutron diffraction and electron microscopy. The data is comprehensive and solid, and the manuscript is suitable to the journal. However, I can not recommend it in current format. One of the major problem is the presentation styles and language. The paper is very hard to read and to be understood, I will expect that authors would lost too.
As a research paper, I would suggest the authors to rewrite the paper using structures as a general paper with sections: introduction, materials and methods, results, discussions and conclusions. Please keep in mind that you have clear picture to show authors your research aim, how you approach, what is new in your study, and what conclusion you draw.
The new main chapter “results and discussion” and “Summary and Conclusion” were created. The paper consists now of the main chapters: Introduction, Materials and Methods, Results and Discussion, Summary and Conclusion. This structure also commonly used in other papers and is most suitable way to present the results in this work in the authors’ opinion.
In order to make the paper more clear some paragraphs have been re-written or moved from the chapter Materials and Methods to Results and Discussion.
For the language, because the problem exist in the entire paper. I mention here only one paragraph (conclusion parts) here for example: the entire conclusion does not really have conclusive data or results, in fact it is more like a summary. Additionally, some word like “therefore” and “furthermore” are misused in the paragraph.
The chapter was renamed to “Summary and Conclusion”. Furthermore, the spelling and grammar was checked in the whole manuscript.
I believe that the entire paper need to be rewritten to give a clear story.
The authors changed the paper accordingly to your suggestions to make the paper more suitable for publication.

Reviewer 2 Report
In the present article the Authors investigated the deformation mechanisms in Ni-base superalloy using combination of two analytical methods: in-situ neutron diffraction and electron microscopy. O found this article very interesting, well organized and surely worth publishing in Metals. However, during reading a few minor comments arises as follows:
- Chapter “Materials and Methods”: in general in this section a detailed description about the investigated materials, performed experiments, analytical methods etc. should be given. Meanwhile, in the present article such descriptions are mixed-up with the results, like e.g. values of volume fractions of phases, stress strain curves etc. Such values should be given in section entitled “Results”, which is missing in the paper.
- Pages 2 and 3, lines 90-93: The Authors declared the values of volume fractions of phases present in the material in the heat-treated condition. However, no single description how exactly these values were calculated. Moreover, no single image showing the microstructure of heat-treated material is shown here (I suspect, that such image is the basis of volume fraction calculation). I recommend to add such image and show the values in the Results section.
- Page 3, Table 1: What kind of chemical composition is it? Nominal or measured? If nominal, it usually is given as a range of certain element content. If it is measured, please clearly define, how the chemical composition was determined.
- Pages 2 and 3, lines 129-130: The Authors wrote that: “The focus is to monitor phase transformations, the evolution of size and volume fractions of precipitates simultaneously.” Please explain in more detail how exactly all of these characteristics (phase transformations, size of precipitates and their volume fraction) are measured/calculated?
- Page 5: Chapter entitled “Results” is missing. I recommend to introduce chapter 3 Results and the following subchapters make as 3.1. In-situ neutron diffraction studies, 3.2. … etc. It would make the readers well guided through the article. However, this is only my suggestion.
- Page 7, line 206 (end of figure 4 caption): an unidentified symbol is introduced, which, I believe, should be “σ=0MPa”. Please correct.
- Page 11, Figure 7: Why two different modes were used for illustration of the undeformed (BSE mode, Fig. 7a) and deformed (SE mode, Fig. 7b) alloy microstructure? It makes difficult to directly compare the differences in the alloy microstructure based on signals coming from different depth of investigated samples. Using one mode (preferentially BSE) is recommended.
- Page 12, line 330: the Authors presented reaults of “Schmid factor”. Please explain what kind of information are hidden behing this factor. I understand that for experts in the field it is obvius, however, please keep in mind, that your publication will be freely available for all kind of readers, even these out of the field.
- Page 15, line 388: The most probably there is a mistake in table caption: there should be “Table 4” instead of “Table 3”. Please correct this.
Despite my few comments, I think that this is a nice article, which can be published in Metals, however, after minor changes according to the comments. Thus I recommend publication after minor revision.
Author Response
Dear reviewer,
Thank you very much for your valuable comments and suggestions on our paper. We followed all of your suggestions/comments and revised the paper accordingly. Below you will find the detailed response to your comments.
Yours sincerely,
Frank Kümmel , Andreas Kirchmayer , Cecilia Solís , Michael Hofmann , Steffen Neumeier and Ralph Gilles
In the present article the Authors investigated the deformation mechanisms in Ni-base superalloy using combination of two analytical methods: in-situ neutron diffraction and electron microscopy. O found this article very interesting, well organized and surely worth publishing in Metals. However, during reading a few minor comments arises as follows:
1. Chapter “Materials and Methods”: in general in this section a detailed description about the investigated materials, performed experiments, analytical methods etc. should be given. Meanwhile, in the present article such descriptions are mixed-up with the results, like e.g. values of volume fractions of phases, stress strain curves etc. Such values should be given in section entitled “Results”, which is missing in the paper.
A new “Results and Discussion” section has been included.
The values of volume fractions of the phases and particle size were determined in a previous work. These results will be published in a future paper. This has been made clearer in the current paper.
The stress/strain curve was moved to the chapter Results and Discussion.
2. Pages 2 and 3, lines 90-93: The Authors declared the values of volume fractions of phases present in the material in the heat-treated condition. However, no single description how exactly these values were calculated. Moreover, no single image showing the microstructure of heat-treated material is shown here (I suspect, that such image is the basis of volume fraction calculation). I recommend to add such image and show the values in the Results section.
See comment above. These values were determined by Neutron Diffraction, Small Angle Neutron Scattering, Atom Probe, and Scanning Electron Microscopy and not by the experiments in this work. These results will be published in a future paper.
3. Page 3, Table 1: What kind of chemical composition is it? Nominal or measured? If nominal, it usually is given as a range of certain element content. If it is measured, please clearly define, how the chemical composition was determined.
The results were measured by spark spectrometry. This was added to the table.
4. Pages 2 and 3, lines 129-130: The Authors wrote that: “The focus is to monitor phase transformations, the evolution of size and volume fractions of precipitates simultaneously.” Please explain in more detail how exactly all of these characteristics (phase transformations, size of precipitates and their volume fraction) are measured/calculated?
The phase transformations were be studied by neutron diffraction and the size and volume fraction of the nano-sized g’ precipitates by small angle neutron scattering. This has been made clearer in the current paper.
5. Page 5: Chapter entitled “Results” is missing. I recommend to introduce chapter 3 Results and the following subchapters make as 3.1. In-situ neutron diffraction studies, 3.2. … etc. It would make the readers well guided through the article. However, this is only my suggestion.
A new main chapter “results and discussion” was implemented. The paper consists now of the main chapters: Introduction, Materials and Methods, Results and Discussion, Summary and Conclusion. This structure also commonly used in other papers and is most suitable way to present the results in this work in the authors’ opinion.
6. Page 7, line 206 (end of figure 4 caption): an unidentified symbol is introduced, which, I believe, should be “σ=0MPa”. Please correct.
The symbol was corrected.
7. Page 11, Figure 7: Why two different modes were used for illustration of the undeformed (BSE mode, Fig. 7a) and deformed (SE mode, Fig. 7b) alloy microstructure? It makes difficult to directly compare the differences in the alloy microstructure based on signals coming from different depth of investigated samples. Using one mode (preferentially BSE) is recommended.
We apologize, this was just a typo. Figure 7b was also taken in BSE mode. Additionally, we realized that some figures shown in the manuscript are taken from the samples deformed at room temperature. We revised the captions accordingly.
8. Page 12, line 330: the Authors presented reaults of “Schmid factor”. Please explain what kind of information are hidden behing this factor. I understand that for experts in the field it is obvius, however, please keep in mind, that your publication will be freely available for all kind of readers, even these out of the field.
The following sentences were added to the manuscript to clarify this point:
Macroscopic plastic deformation begins as soon as the applied stress exceeds the yield point of the material and massive dislocation movement takes place. However, the plastic deformation is not homogeneously distributed in polycrystalline materials. In these materials, the direction of the individual grains in relation to the direction of loading must be taken into account. A dislocation moves as a result of a force that acts on it in the slip plane in the direction of the Burgers vector (slip direction). Therefore, the resulting shear stress t in the sliding system is relevant and not the applied tensile stress s. The resulting shear stress t is a function of the tensile stress s, the angle between the tensile direction and direction normal to the glide plane j and the angle between the tensile direction and the direction of the glide plane w, equation (4). The angle dependence can be summarized by the Schmid factor S.
τ=σ ∙ cos φ ∙ cos ω = σ ∙ S (4)
9. Page 15, line 388: The most probably there is a mistake in table caption: there should be “Table 4” instead of “Table 3”. Please correct this.
The labeling was corrected.
Despite my few comments, I think that this is a nice article, which can be published in Metals, however, after minor changes according to the comments. Thus I recommend publication after minor revision.
The authors changed the paper accordingly to your suggestions to make the paper more suitable for publication.

Round 2
Reviewer 1 Report
The authors have answered my previous questions
Reviewer 2 Report
The Authors seriously reacted on Revoewers' comments and improved the quality of the article. Thus I recommend to accept the article in the present form.